# 2-[^18^F]FDG PET/CT as a Predictor of Microvascular Invasion and High Histological Grade in Patients with Hepatocellular Carcinoma

**DOI:** 10.3390/cancers13112554

**Published:** 2021-05-23

**Authors:** Aida Sabaté-Llobera, Judit Mestres-Martí, Gabriel Reynés-Llompart, Laura Lladó, Kristel Mils, Teresa Serrano, Montserrat Cortés-Romera, Esther Bertran, Isabel Fabregat, Emilio Ramos

**Affiliations:** 1PET Unit, Department of Nuclear Medicine-IDI, University Hospital of Bellvitge, L’Hospitalet de Llobregat, 08907 Barcelona, Spain; jmestres@clinic.cat (J.M.-M.); greynes@iconcologia.net (G.R.-L.); montserrat.cortes.idi@gencat.cat (M.C.-R.); 2Bellvitge Biomedical Research Institute (IDIBELL), L’Hospitalet de Llobregat, 08908 Barcelona, Spain; laurallado@bellvitgehospital.cat (L.L.); kristel.mils@bellvitgehospital.cat (K.M.); tserrano@bellvitgehospital.cat (T.S.); ifabregat@idibell.cat (I.F.); eramos@bellvitgehospital.cat (E.R.); 3Department of Medical Physics, Catalan Institute of Oncology, Duran i Reynals Hospital, L’Hospitalet de Llobregat, 08908 Barcelona, Spain; 4Faculty of Medicine and Health Sciences, University of Barcelona, L’Hospitalet de Llobregat, 08907 Barcelona, Spain; 5Hepato-Biliary, Pancreatic and Liver Transplantation Unit, Department of Surgery, University Hospital of Bellvitge, L’Hospitalet de Llobregat, 08907 Barcelona, Spain; 6Department of Pathology, University Hospital of Bellvitge, L’Hospitalet de Llobregat, 08907 Barcelona, Spain; 7Oncology Program, Centro de Investigación Biomédica en Red, Enfermedades Hepáticas y Digestivas (CIBEREHD), National Biomedical Research Institute on Liver and Gastrointestinal Diseases, Instituto de Salud Carlos III, 28029 Madrid, Spain; ebertran@idibell.cat; 8TGF-ß and Cancer Group, Oncobell Program, Bellvitge Biomedical Research Institute (IDIBELL), L’Hospitalet de Llobregat, 08908 Barcelona, Spain

**Keywords:** hepatocellular carcinoma, 2-[^18^F]FDG, PET/CT, liver resection, standardized uptake value, lean body mass, microvascular invasion, cellular differentiation.

## Abstract

**Simple Summary:**

In recent years, functional imaging techniques have been increasingly studied to preoperatively identify the aggressive features in hepatocellular carcinoma (HCC). PET/CT with 2-deoxy-2-[^18^F]fluoro-d-glucose (FDG) is not routinely used for the diagnosis and staging of an HCC due to its low uptake of this radiopharmaceutical especially in well-differentiated lesions. However, FDG uptake in an HCC seems to relate to biological aggressiveness, being able to predict certain factors such as microvascular invasion. The present work aimed to assess the prognostic value of performing a baseline PET/CT with FDG in patients with an HCC who were subsequently treated with a tumor resection, trying to identify which metabolic parameters may predict the presence of histological factors of a poor prognosis. In our series, an increased SULpeak of the ratio tumor/liver 60 min after an FDG injection (TLRpeak60) seems a promising parameter to predict histological factors of a poor prognosis that could aid decision-making in this group of patients.

**Abstract:**

Hepatocellular carcinoma (HCC) generally presents a low avidity for 2-deoxy-2-[^18^F]fluoro-d-glucose (FDG) in PET/CT although an increased FDG uptake seems to relate to more aggressive biological factors. To define the prognostic value of PET/CT with FDG in patients with an HCC scheduled for a tumor resection, forty-one patients were prospectively studied. The histological factors of a poor prognosis were determined and FDG uptake in the HCC lesions was analyzed semi-quantitatively (lean body mass-corrected standardized uptake value (SUL) and tumor-to-liver ratio (TLR) at different time points). The PET metabolic parameters were related to the histological characteristics of the resected tumors and to the evolution of patients. Microvascular invasion (MVI) and a poor grade of differentiation were significantly related to a worse prognosis. The SULpeak of the lesion 60 min post-FDG injection was the best parameter to predict MVI while the SULpeak of the TLR at 60 min was better for a poor differentiation. Moreover, the latter parameter was also the best preoperative variable available to predict any of these two histological factors. Patients with an increased TLRpeak60 presented a significantly higher incidence of poor prognostic factors than the rest (75% vs. 28.6%, *p* = 0.005) and a significantly higher incidence of recurrence at 12 months (38% vs. 0%, *p* = 0.014). Therefore, a semi-quantitative analysis of certain metabolic parameters on PET/CT can help identify, preoperatively, patients with histological factors of a poor prognosis, allowing an adjustment of the therapeutic strategy for those patients with a higher risk of an early recurrence.

## 1. Introduction

Hepatocellular carcinoma (HCC) is the most common primary cancer of the liver with estimates of it being the seventh most incident cancer type and the third with the highest mortality in the world in 2020 [1]. Among the curative treatments, surgical resection is preferred in patients without cirrhosis as well as in those without clinically significant portal hypertension and Child–Pugh class A [2]. However, the risk of recurrence after a resection is high (about 70% at five years) [3] with the grade of differentiation, the presence of vascular invasion and the existence of satellite nodules the most important prognostic factors for an early recurrence [4,5]. Predicting the existence of these factors would allow a more patient-specific treatment with a better selection of those individuals who could benefit from surgery and with the highest probability of a long-term survival. 

Even if histological confirmation was mandatory before treatment of most neoplasms, it is usually not required and even discouraged as a routine procedure in an HCC mainly due to the risk of bleeding, tumor seeding and the possibility of false negative results [2,6]. Therefore, in most cases, the diagnosis is done with a multiphasic contrast-enhanced CT or MRI. In the last years, different studies have evaluated the capacity of conventional and functional imaging methods to preoperatively identify aggressive features of an HCC especially microvascular invasion (MVI) [7,8]. Unlike in other malignancies, PET/CT with 2-deoxy-2-[18F]fluoro-d-glucose (FDG) is not routinely used as part of the diagnostic and staging workup of an HCC as it shows a low uptake of this radiopharmaceutical especially in well-differentiated lesions [9]. However, FDG uptake in HCC lesions seems to relate to more aggressive biological factors such as higher alpha-fetoprotein levels and to predict MVI and an early recurrence after a curative treatment [10,11,12].

In this scenario, we aimed to assess the prognostic value of performing a baseline PET/CT with FDG in patients diagnosed with an HCC and subsequently treated with a tumor resection, specifically trying to identify those metabolic parameters associated with the histological factors of a high risk of recurrence.

## 2. Materials and Methods

This study was approved by the Clinical Research Ethics Committee of our institution (code PR191/17). It included patients diagnosed with an HCC of stages BCLC 0 and A, operated on between June 2018 and November 2020 and with an 18F-FDG PET/CT for the purpose of the study during the 12 weeks prior to the surgery. Clinical, imaging and histological data were collected prospectively and retrieved from our database for the analysis. 

### 2.1. Patient Selection

Patients included in this study were diagnosed with an HCC according to the criteria of the established guidelines [13] and evaluated by the multidisciplinary team for liver tumors of our institution who considered them as candidates for surgical treatment. They then followed the standard workup for surgical patients with the additional performance of a PET/CT with FDG. 

The exclusion criteria for the participation in the present study were: (a) radiological evidence of more than one HCC nodule; (b) impossibility of performing a PET/CT; (c) the negation of the patient to be part of the study; (d) previous or simultaneous diagnosis of another neoplasm; (e) previous treatment of another HCC nodule.

### 2.2. PET/CT Acquisition

The PET/CT studies were performed in a Discovery IQ 5-ring (GE Healthcare, Waukesha, WI). The patients fasted for at least 6 h and were instructed to drink water willingly. All of them had blood glucose levels below 11 mmol/L before the FDG injection and were administered 2.7 MBq/kg. Images were acquired at 62 ± 5 min after the radiotracer administration from mid-thigh to the skull base with arms raised. The PET images were set at 2 min/bed position and reconstructed using an OSEM iterative reconstruction with modeling PSF (12 subsets and 4 iterations) as detailed in our previous work [14]. Delayed images of the liver were acquired at 90, 120 and 180 min after the radiotracer administration. A low dose, non-contrast enhanced CT was performed as part of the PET/CT study used both for attenuation correction and for anatomic localization (140 kV, modulated intensity, slice thickness of 2.5 mm). All patients had a diagnostic contrast-enhanced three-phase CT study before the PET/CT. However, in those cases in which this CT was performed more than four weeks before the PET/CT, an additional contrast-enhanced CT of the liver in the arterial and portal phases (32-slices scanner, 120 kV with a reconstructed slice of 1.25 mm and automatic exposure control) was performed for a more accurate localization of the HCC lesion. 

### 2.3. Image Evaluation

A first qualitative assessment was performed, visually determining if the HCC lesions were positive (FDG uptake higher than the surrounding liver parenchyma) or negative (FDG uptake equal to or below liver activity). For the semi-quantitative analysis, an automatic spherical region of interest (VOI) was placed over the liver lesion with a minimum volume of 20 cm^3^. From this VOI, the maximum, mean and peak standardized uptake values corrected for lean body mass (SULmax, SULmean and SULpeak) were obtained. Additionally, two automatic VOIs of 20 cm^3^ were also placed over the non-cancerous liver tissue, preferably one on the left hepatic lobe and one on the posterior segments of the right hepatic lobe. The average of the SUL values obtained from these VOIs was used as background liver activity to calculate the tumor-to-liver ratio (TLR).

Therefore, the metabolic parameters obtained were SULmax, SULmean, SULpeak, TLRmax, TLRmean and TLRpeak at 60, 90, 120 and 180 min after the injection of the radiopharmaceutical.

### 2.4. Pathological Study of the Resected Lesions 

The gross inspection of the surgery specimens included serial sections of less than 1 cm thick in order to properly identify the number of nodules, their sizes and their location, trying to correlate these findings with the radiological ones. The presence of satellite nodules, MVI and a margin invasion as well as the differentiation grade (G1–G4, according to Edmondson and Steiner) were assessed by microscopy.

### 2.5. Post-Surgical Follow-up

Patients did not receive any adjuvant treatment. They were visited one month after they were discharged from the hospital (post-surgery stay) and then periodically every three months for the first year and every six months from then on. A follow-up CT or MRI was performed every six months. An HCC recurrence was considered when there was a radiological finding suggestive of a tumor relapse (intra-hepatic and/or extra-hepatic).

### 2.6. Statistical Analysis

Data were analyzed in December 2020. The statistical analysis was performed using SPSS version 25 (Chicago, IL, USA). The continuous variables were presented as means with standard deviations and compared using the Mann–Whitney U test. Categorical data were shown as frequencies (%) and compared using Fisher’s exact test. Univariate and multivariate analyses were performed to assess the role of clinical, histological and metabolic parameters obtained at the PET/CT in predicting the presence of factors associated with a worse outcome (Kaplan–Meier test and a Cox regression model). Only variables with *p* < 0.05 at the univariate analyses were included in the multivariate ones. To identify the quantitative metabolic variables associated with histological factors of a poor prognosis, the Mann–Whitney U test for two independent groups was applied. Once identified, cut-off values at percentiles 25, 50 and 75 (p25, p50 and p75, respectively) of these quantitative variables were determined to convert them into qualitative variables. For all of these new qualitative variables, a ROC curve was obtained and the area under the curve (AUC) was calculated as an indicator of the predictive value for the presence of the histological factors of a poor prognosis. The Kaplan–Meier curves were used to depict the recurrence and were compared using the log-rank test. 

## 3. Results

During the period of the study, 67 consecutive resections for an HCC were performed. Forty-one of these patients had a PET/CT before the surgery and had given their consent to participate in the study and constituted the study population. 

At the diagnosis, 35 patients (85.4%) were asymptomatic. Most of the patients (61%) had a chronic hepatopathy, all of them with a Child–Pugh score A. In 15 patients (36.6%), liver disease was virus related, mostly due to HCV—in two cases, with an active HCV infection—patients with an HBV related disease did not show viral replication. The cross-sectional imaging preoperative study revealed a mean tumor size of 42 ± 22 mm and did not identify multiple liver lesions in any of the cases. In five patients (12.2%), a liver biopsy was performed before the surgery and two patients underwent a transarterial chemoembolization (TACE) after the PET/CT and before the tumor resection. At the post-surgical evaluation, two patients were scheduled for a liver transplantation due to the histological characteristics of a poor prognosis at the resection specimen and in five patients (12.2%), additional HCC nodules not identified at the imaging tests were demonstrated. There was no post-operative mortality. 

The epidemiological, clinical and histological characteristics of the study population are detailed in Table 1. 

### 3.1. Recurrence and Survival Data

After a mean follow-up of 8.0 ± 5.2 months, seven patients (17.1%) were diagnosed with an HCC recurrence. All of them had a non-local liver relapse and in one case, disease outside the liver was also detected. Among these patients, three received a local treatment (one with TACE and two with radiofrequency), two started treatment with sorafenib and the other one was scheduled for a liver transplantation but died before the transplantation due to the disease progression. The remaining patient was a recent relapse and was pending a therapeutic decision. 

The overall recurrence at 12 months from the surgery was 29.3%. At the end of the follow-up period, there were three deceased patients (7.3%), two of them due to tumor related causes.

### 3.2. Clinical and Histological Factors Predicting Tumor Recurrence after the Surgery

The presence of MVI and a poor grade of differentiation as well as a previous HBV infection were significantly related to a worse outcome at the univariate analysis. At the multivariate analysis, none of these variables were significant although the small sample size and the short follow-up period could have influenced the results. However, a poor cellular differentiation was close to significance (Table 2). Due to the small number of patients with a previous HBV infection, only a poor cellular differentiation and MVI were selected to continue the study. 

### 3.3. Correlation between Metabolic Parameters on PET/CT and Histological Factors of a Poor Prognosis

#### 3.3.1. Presence of MVI

To identify those metabolic variables predictive of MVI, the metabolic values of the different variables obtained in patients with and without MVI were compared using the Mann–Whitney U test. A qualitative conversion of those with significant differences between patients with and without MVI and without missing values (SULmax, SULpeak, TLRmax, TLRmean and TLRpeak; all at the 60 min images) was performed using p25, p50 and p75 as cut-off values.

In the majority of the cases using a cut-off value in the p50 or p75, a significant relationship between the metabolic activity of the tumor and the presence of MVI was observed. The AUC was calculated for each of these metabolic parameters, being the SULpeak of the lesion at 60 min (SULpeak60) with a cut-off value of 2.26 (p50) the one showing the best results (AUC 0.716). Patients with an SULpeak60 ≥ 2.26 presented bigger HCC lesions and had a greater incidence of MVI (Table 3). 

#### 3.3.2. Poor Cellular Differentiation

In the same way reported for MVI, to identify those metabolic parameters predictive of a poor cellular differentiation, a comparison of the results obtained for each metabolic variable in patients with and without poorly differentiated tumors (G3 and G4 vs. G1 and G2) was performed using the Mann–Whitney U test. Five metabolic parameters, corresponding with the same ones as previous (SULmax, SULpeak, TLRmax, TLRmean and TLRpeak; all at the 60 min images) were qualitatively converted using p25, p50 and p75 as cut-off values. 

In all of the five variables using a cut-off value in the p50 or p75, a significant relationship between the metabolic activity of the tumor and the grade of differentiation was observed. As in the previous section, the AUC was also calculated for each of these variables, now being SULpeak of the TLR at 60 min (TLRpeak60) with a cut-off value of 1.20 (p50) the one with the best performance (AUC 0.744). 

Patients with a TLRpeak60 ≥ 1.20 had a higher incidence of a poor cellular differentiation, a higher presence of satellite nodules, a greater incidence of MVI and bigger HCC lesions (Table 4).

### 3.4. Presence of MVI and/or Poor Cellular Differentiation

For this analysis, only the p50 values of SULpeak60 and TLRpeak60 were used. Proceeding as explained in the previous section, the AUC for both parameters was calculated with TLRpeak60 obtaining the best results (AUC 0.732). Patients with TLRpeak60 ≥ 1.20 had MVI or a poor cellular differentiation in 75% of the cases while these factors were present in only 28.6% of the remaining study population (*p* = 0.005). This was the only variable available before the surgery that predicted the presence of any of the two histological factors of a poor prognosis (Table 5). 

Furthermore, patients with increased values of TLRpeak60 presented a significantly higher incidence of disease recurrence at 12 months compared with the rest of the patients (38% vs. 0%, *p* = 0.014). No significant differences regarding survival at 12 months were observed. 

## 4. Discussion

Beyond the conventional criteria used to select the best candidates for an HCC resection, having presurgical parameters to identify those patients at a higher risk of recurrence could be useful to offer a more personalized management of the oncological process. Lately, a few studies have defined patterns that could determine MVI on MRI. Mulé et al. [15] studied the degree of gadobenate dimeglubine enhancement in the tumor during the delayed hepatobiliary phase and concluded that a lesion-to-liver contrast enhancement ratio of −4.72% or less could accurately predict MVI and moderate-poor differentiation grade in an HCC while the work by Ahn et al. [16] revealed that the peritumoral enhancement of gadoxetic acid may suggest MVI. However, conventional imaging techniques (CT and MRI) still face several difficulties in predicting tumor biology although the recent introduction of a radiomics evaluation on MRI could help overcome a few of these limitations [17]. In this fashion, a recently published systematic review and meta-analysis [8] compared radiomics and non-radiomics features in different imaging techniques to predict MVI before an HCC surgery. Despite no significant differences being observed between both groups, they concluded that it was more desirable to conduct the radiomics analysis due to its objectivity of quantified features and the high consistency among the studies.

Even if an HCC is known to have a rather low FDG uptake due to the high activity of glucose-6-phosphatase [18], several works have proved that HCC lesions with a higher metabolic activity tend to present more aggressive biological factors related to a worse prognosis [9,12,19]. The current study supports the previously reported results on PET/CT with FDG and defined the most reliable metabolic parameters to better predict the histological factors of a poor prognosis in our population (MVI and a poor grade of cellular differentiation). 

In order to determine these parameters, we performed a semi-quantitative analysis of several PET metabolic variables at different time points after an FDG injection. As the liver is a central metabolism-regulating organ, patients with impaired liver function often present malnutrition in the form of obesity, sarcopenia or sarcopenic obesity [20]. Therefore, instead of analyzing the regular PET metabolic parameters (SUV), which are based on the individual’s body weight, we obtained the metabolic parameters corrected by the lean body mass of the patients (SUL), which have been reported to be more adequate to quantify the FDG uptake especially in obese patients [21,22,23]. To normalize the variations of glucose metabolism due to this possibly altered function in the liver, the ratio between the FDG uptake in the HCC lesion and in the non-tumoral liver parenchyma (TLR) was also calculated.

Different studies have reported higher SUVmax and TLRmax values associated with a higher risk of MVI and a poorer cellular differentiation [11,12,24]. A recent review by Lee et al. [25] analyzed, among other aspects, the prognostic value of PET/CT with FDG in an HCC. In many of the included publications, PET images were solely evaluated qualitatively (the presence or absence of FDG uptake in the lesion) and in those cases in which a quantitative parameter was studied, TLR was the most frequently chosen as it was better correlated with the HCC doubling time and it represented the metabolic activities of the lesion more precisely than SUV. In our study, SULpeak and TLRpeak were stronger predictors of a poor prognosis than SULmax and TLRmax even if the latter were also related to MVI and the grade of differentiation. To the best of our knowledge, no previous studies have checked the value of SUV/SULpeak in an HCC as SUVmax is the standard semi-quantitative parameter of routine use. SULpeak appears to be less affected by image noise than SUV/SULmax, typically gaining precision [26], so is preferred in the evaluation of the response to the treatment of solid tumors. However, the maximum and peak metrics are highly correlated among them so the routine checking of SUV/SULpeak in oncologic studies seems unnecessary unless a foreseeable treatment response evaluation PET/CT is expected.

TLRpeak60 was the only independent factor available before the surgery that could predict either MVI or a poor cellular differentiation (Figure 1). This information could be of paramount importance, as it could change the management of patients: patients with liver cirrhosis and an increased TLRpeak60 would have a higher risk of histological factors of a poor prognosis so they could be directly selected for liver transplantation instead of liver resection as, according to a few authors [27], liver transplantation should be offered to patients showing MVI and/or satellite nodules in the resected specimen. Others suggest that a favorable response to TACE allows the identification of those individuals who would more likely result in a better outcome after surgery [28,29]. Therefore, patients with contraindications for liver transplantation and with metabolic data suggesting the presence of histological factors of a poor prognosis could be offered a neoadjuvant treatment with TACE, which would serve as a way of selecting patients before the surgery.

In the present study, the tumor size after liver surgery did not represent a significant prognostic factor for tumor recurrence although HCC lesions ≥ 30 mm at the pathological study tended to show a higher SULpeak60 and TLRpeak60 on PET/CT. On cross-sectional imaging, patients with MVI usually had bigger tumors but only those with advanced fibrosis and HCC lesions of 30 mm or larger were significantly associated with MVI (*p* = 0.018) (Figure 2). In their work with small HCC lesions (≤30 mm), Kobayashi et al. [30] found SUVmax ≥ 3.2 to be an independent predictor of MVI with an AUC of 0.712. However, its positive predictive value was only 35% so they did not recommend the use of this parameter as a single MVI predictor to select patients for a liver resection or transplantation. In any case, studying the relationship between the tumor size and MVI or the grade of differentiation was not a scope of our work even if previous studies [11,12] found the tumor size to be a significant predictor of MVI, which could be somehow explained by the greater proportion of patients with advanced fibrosis compared with our series. Contradictory results have been published on the prognostic significance of the tumor size [11,12,16,31] so further research to draw more solid conclusions on this factor is granted.

Having a history of HBV infection appeared to be a significant predictor of disease recurrence after a liver resection at the univariate analysis. However, only 17.1% of patients had positivity for the surface HBV antigen—all of them with negative DNA—and at the multivariate analysis, this factor was much less significant than MVI and the grade of differentiation so it was not further studied. However, 71% of these patients had an increase in both SULpeak60 and TLRpeak60, which would suggest that HBV infection, even in inactive carriers, was related to high risk histological factors (Figure 3). In the study by Jeng et al. [32] including 48 HBV carriers with liver lesions, 20 out of 36 HCCs showed FDG uptake compared with none of the twelve benign lesions. Nevertheless, additional factors influencing FDG uptake were not studied and the number of patients was small so more extensive research on this topic could be interesting to better define its prognostic meaning. 

FDG uptake has been correlated with the overall survival and/or progression-free survival in an HCC in various stages and after different treatment approaches [33]. Cho et al. [10] reported that patients with a higher SUVmax on an HCC were more prone to relapse after a liver resection although differences between the groups were not significant. Similar results were obtained in the present study as even if TLRpeak60 was not a significant predictor of overall survival or disease-free survival, patients with higher values showed a higher recurrence at 12 months. Not all of the previous studies on this topic have been able to confirm a relationship between PET/CT findings and the prognosis. This could be due to the specific scenario where the study was performed. As reported in the review by Lee et al. [25], in different publications by the Korean Society of Nuclear Medicine Clinical Trial Network (KSNMCTN), TLR appeared as an independent predictor of recurrence and survival in a cohort of patients treated with a curative intention but not in patients who underwent a palliative treatment with TACE in which liver function is a more relevant factor. In our case, increasing the study population and the follow-up time would probably better define the role of PET/CT and of the different metabolic parameters in recurrence and survival. 

The major strength of this study was the prospective recruitment of patients according to strict criteria of surgery selection; to the best of our knowledge, only one previously published work [12] was designed in a prospective way. PET/CT studies were also performed and analyzed following a homogeneous procedure to obtain reliable semi-quantitative parameters. The definition of the cut-off values to predict histological high risk factors is an important issue as it may help in taking decisions regarding the management of each patient. Despite this, as the number of patients was relatively small, increasing the study population could lead to slightly different results regarding the significance of the semi-quantitative parameters on PET imaging and their cut-off values, which was certainly a limitation of the current work. Moreover, the short follow-up time limited a proper evaluation of PET/CT on survival so an update of the results after a longer period of time would bring more precise information on this point.

## 5. Conclusions

A semi-quantitative evaluation of several metabolic parameters obtained in a PET/CT with FDG allowed the identification of patients with a higher risk of MVI or a poor cellular differentiation. The routine performance of a PET/CT previous to an HCC resection could be useful to obtain a better understanding of the biological aggressiveness of the neoplastic disease in each individual. 

## Figures and Tables

**Figure 1 cancers-13-02554-f001:**
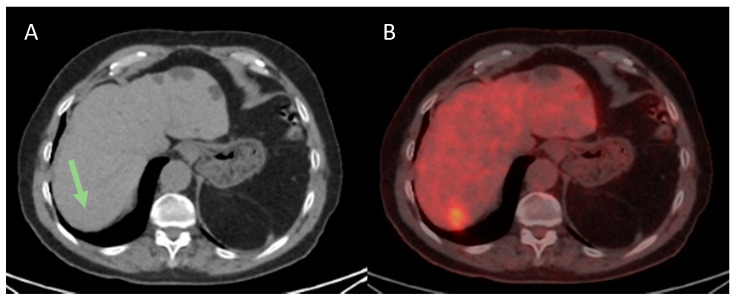
Hepatocellular carcinoma (HCC) of 33 mm, difficult to identify in the non-contrast enhanced CT ((**A**), green arrow) but with a high FDG uptake (**B**), with SULpeak60 of 3.98 and TLRpeak60 of 1.91. The resection specimen showed a grade 3 HCC with microvascular invasion (MVI). The disease recurred seven months after surgery.

**Figure 2 cancers-13-02554-f002:**
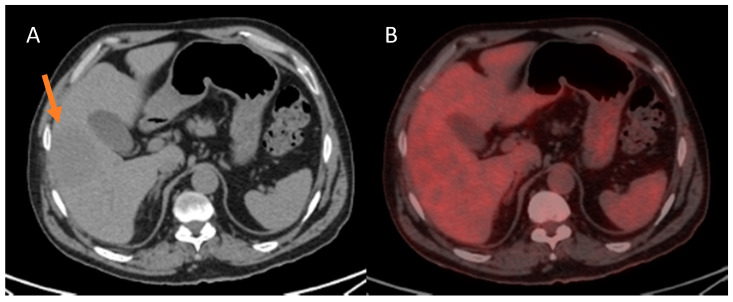
Patient without liver disease who developed a hepatocellular carcinoma (HCC) of 65 mm on CT ((**A**), orange arrow) showing a similar FDG uptake as the rest of the liver parenchyma (**B**) with SULpeak60 of 2.0 and TLRpeak60 of 1.02. At the pathology analysis, a grade 2 HCC with microvascular invasion (MVI) was confirmed. No disease recurrence was evidenced after a follow-up of nine months.

**Figure 3 cancers-13-02554-f003:**
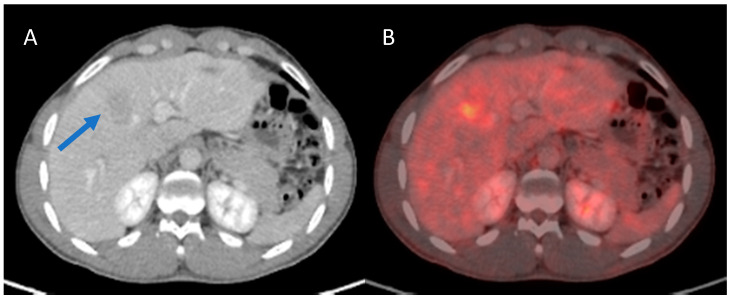
A hepatocellular carcinoma (HCC) of 44 mm was diagnosed in this patient ((**A**), blue arrow) who had a positive surface hepatitis B virus antigen (HBV S Ag+). The lesion showed an intense FDG uptake (**B**) with SULpeak60 of 4.56 and TLRpeak60 of 2.10. The resection specimen evidenced a grade 3 HCC with microvascular invasion (MVI). The disease recurred two months after the surgery.

**Table 1 cancers-13-02554-t001:** Characteristics of the study population.

Variables	*n* = 41
Sex (M/F)	39/2
Age (years), mean (SD)	61 (11.4)
BMI (kg/m^2^), mean (SD)	27.1 (4.3)
HVPG ≥10 mmHg (measured in 10 patients), n	1
MELD score, mean (SD)	7.2 (1.5)
Liver disease, n	
HBV related	7
HCV related	8
Alcohol related	9
NASH	1
None or unknown	16
HCV Ab+, n (%)	12 (29.3)
HCV RNA+, n	2
HBV S Ag+, n (%)	7 (17.1)
HBV DNA+, n	0
AFP (ng/mL), mean (SD)	1573.2 (926.1)
**Tumor characteristics on cross-sectional imaging**	
Size of the largest nodule (mm), mean (SD)	42 (22)
**Liver and kidney function test results**	
Total bilirubin (µmol/L), mean (SD)	10.0 (4.3)
Albumin (g/L), mean (SD)	45.9 (3.5)
Platelet (×10^5^/mm^3^), mean (SD)	209 (68)
Time of prothrombin ratio, mean (SD)	1.11 (0.43)
Creatinine (µmol/L), mean (SD)	81.6 (20.0)
ASA classification (I/II/III/IV)	1/19/21/0
MELD, mean (SD)	7.2 (1.5)
Child–Pugh	5 (0)
**Intra-operative and post-operative variables**	
Open/laparoscopic	21/20
Major resection, n (%)	3 (7.3)
Anatomical resection, n (%)	17 (41.5)
Pringle maneuver yes/no	37/4
Perioperative transfusion yes/no	4/37
Post-operative complications, Clavien classification (No/I/II/IIIa)	28/6/4/3
Mortality, n	0
Hospitalization time (days), mean (SD)	6.9 (6.8)
**Pathological variables**	
Fibrosis grade (0/F1 + F2/F3 + F4)	13/6/22
Histological type (HCC/hepatocholangiocarcinoma)	39/2
Number of tumors, mean (SD)	1.12 (0.33)
Multiple tumors, n (%)	5 (12.2)
Tumor size (mm), mean (SD)	43 (24.8)
Grade of differentiation (G1 + G2/G3 + G4)	29/12
Satellite nodules, n (%)	4 (9.8)
Microvascular invasion, n (%)	17 (41.5)
Margin invasion, n (%)	6 (14.6)

HVPG: hepatic venous pressure gradient; MELD: model for end-stage liver disease; HBV: hepatitis B virus; HCV: hepatitis C virus; NASH: non-alcoholic steatohepatitis; AFP: alpha-fetoprotein; ASA: American Society of Anesthesiologists; HCC: hepatocellular carcinoma.

**Table 2 cancers-13-02554-t002:** Analysis of prognostic factors for tumor recurrence after a liver resection.

	Univariate Analysis	Multivariate Analysis	
Preoperative Variables	*p*-Value	HR 95% CI	*p*-Value
Sex (M/F)	0.860		
Age ≥ 70 years	0.076		
BMI ≥ 30 kg/m^2^	0.582		
MELD score ≥ 9	0.418		
ASA classification (I/II/III/IV)	0.177		
Liver stiffness (>14 kPa)	0.715		
HBV S Ag+	0.009	1.18 (0.22–6.31)	0.844
HCV Ab+	0.826		
AFP ≥ 200 ng/mL	0.563		
**Intra-operative and post-operative variables**			
Open/laparoscopic	0.684		
Major resection	0.380		
Pringle maneuver	0.065		
Anatomical resection	0.187		
Perioperative transfusion	0.372		
Post-operative complications	0.235		
**Histopathological variables**			
Fibrosis grade (0 + F1 + F2/F3 + F4)	0.641		
Multiple nodules	0.119		
Satellite nodules	0.280		
Microvascular invasion (MVI)	0.014	3.93 (0.39–39.21)	0.243
Differentiation grade (G1 + G2/G3 + G4)	0.001	8.51 (0.85–84.75)	0.068
Histology (HCC/hepatocholangiocarcinoma)	0.385		
R1 resection	0.591		

BMI: body mass index; MELD: model for end-stage liver disease; ASA: American Society of Anesthesiologists; HBV: hepatitis B virus; HCV: hepatitis C virus; AFP: alpha-fetoprotein; HCC: hepatocellular carcinoma.

**Table 3 cancers-13-02554-t003:** Comparison between the two groups of patients according to the metabolic activity based on SULpeak60 with the cut-off in the p50 (2.26).

Variables	SULpeak60 ≥ 2.26 (n = 21)	SULpeak60 < 2.26 (n = 20)	*p*-Value
Sex (M/F)	21/0	18/2	0.232
Age (years), mean (SD)	60.4 (12.8)	62.6 (10.4)	0.547
BMI (kg/m^2^), mean (SD)	26.6 (4.0)	27.6 (4.6)	0.450
HCV Ab+, n (%) HCV RNA+, n (%)	2 (9.5) 0	10 (50) 2 (10)	0.006 1.000
HBV S Ag+, n (%)	7 (33.3)	0 (0)	0.009
AFP (ng/mL), mean (SD)	2091.0 (13109.7)	24 (66.8)	0.359
AFP ≥ 200 ng/mL, n (%)	3 (14.3)	1 (5.6)	0.609
**Tumor characteristics on cross-sectional imaging**			
Size of the nodule (mm), mean (SD)	5.1 (2.6)	3.2 (1.2)	0.006
**Functional preoperative status**			
ASA classification (I/II/III/IV)	1/6/14/0	0/13/7/0	0.053
MELD	7.19 (1.8)	7.2 (1.2)	0.985
**Intra-operative and post-operative variables**			
Open/laparoscopic	13/8	8/12	0.217
Major resection, n (%)	2 (9.5)	1 (5)	1.000
Anatomical resection, n (%)	10 (47.6)	8 (35)	0.530
Pringle maneuver yes/no	18 (85.7)	19 (95.0)	1.000
Perioperative transfusion yes/no	3/18	1/19	0.606
Post-operative complications. Clavien classification (No/I/II/IIIa)	14/4/1/2	14/2/3/1	0.577
Mortality, n	0	0	
Hospitalization time (days), mean (SD)	6.95 (5.5)	7.1 (8.0)	0.982
**Pathological variables**			
Fibrosis grade (0 + F1 + F2/F3 + F4)	11/10	8/12	0.536
Histological type (HCC/hepatocholangiocarcinoma)	21/0	18/2	0.232
Number of tumors	1.1 (0.4)	1.05 (0.2)	0.174
Multiple tumors, n (%)	4 (19)	1 (5)	0.343
Tumor size (mm), mean (SD)	5.2 (2.8)	3.4 (1.6)	0.013
Tumor size ≥ 30 mm, n (%)	18 (85.7)	10 (50)	0.020
Grade of differentiation (G1 + G2/G3 + G4)	12/9	17/3	0.085
Satellite nodules, n (%)	3 (14.3)	1 (5)	0.606
Microvascular invasion, n (%)	13 (61.9)	4 (20)	0.010
R1 resection, n (%)	5 (23.8)	1 (5.3)	0.186

BMI: body mass index; HCV: hepatitis C virus; HBV: hepatitis B virus; AFP: alpha-fetoprotein; ASA: American Society of Anesthesiologists; MELD: model for end-stage liver disease; HCC: hepatocellular carcinoma.

**Table 4 cancers-13-02554-t004:** Comparison between the two groups of patients according to metabolic activity based on TLRpeak60 with the cut-off in the p50 (1.20).

Variables	TLRpeak60 ≥ 1.20 (*n* = 20)	TLRpeak60 < 1.20 (*n* = 21)	*p*-Value
Sex (M/F)	20/0	19/2	0.488
Age (years), mean (SD)	59.0 (13.5)	63.8 (8.8)	0.183
BMI (kg/m^2^), mean (SD)	26.4 (4.3)	27.8 (4.3)	0.327
HCV Ab+, n (%) HCV RNA+, n (%)	3 (15) 0 (0)	9 (42.9) 2 (22.2)	0.085 1.000
HBV S Ag+, n (%)	5 (25)	2 (9.5)	0.238
AFP (ng/mL), mean (SD)	3046.4 (13,432.9)	22.6 (64.4)	0.327
AFP ≥ 200 ng/mL, n (%)	3 (15)	1 (5.3)	0.605
**Tumor characteristics on cross-sectional imaging**			
Size of the largest nodule (mm), mean (SD)	5.2 (2.7)	3.3 (1.2)	0.008
**Functional preoperative status**			
ASA classification (I/II/III/IV)	1/7/12/0	0/12/9/0	0.210
MELD	7.6 (0.9)	6.8 (1.9)	0.104
**Intra-operative and post-operative variables**			
Open/laparoscopic	11/9	10/11	0.758
Major resection, n (%)	2 (10)	1 (4.8)	0.606
Anatomical resection, n (%)	10 (50)	7 (33.3)	0.350
Pringle maneuver yes/no	17/3	20/1	0.343
Perioperative transfusion yes/no	3/17	1/20	0.343
Post-operative complications. Clavien classification (No/I/II/IIIa)	14/3/1/2	14/3/3/1	0.727
Mortality, n	0	0	
Hospitalization time (days), mean (SD)	6.9 (5.6)	7.0 (7.8)	0.945
**Pathological variables**			
Fibrosis grade (0 + F1 + F2/F3 + F4)	11/9	8/13	0.354
Histological type (HCC/hepatocholangiocarcinoma)	20/0	19/2	0.488
Number of tumors	1.2 (0.4)	1.05 (0.2)	0.151
Multiple tumors, n (%)	4 (20)	1 (4.8)	0.184
Tumor size (mm), mean (SD)	4.9 (2.6)	3.9 (2.3)	0.265
Tumor size > 30 mm, n (%)	18 (90)	10 (47.6)	0.006
Grade of differentiation (G1 + G2/G3 + G4)	10/10	19/2	0.006
Satellite nodules yes/no	4/16	0/21	0.048
Microvascular invasion, n (%)	12 (60)	5 (23.8)	0.028
R1 resection, n (%)	5 (25)	1 (5)	0.182

BMI: body mass index; HCV: hepatitis C virus; HBV: hepatitis B virus; AFP: alpha-fetoprotein; ASA: American Society of Anesthesiologists; MELD: model for end-stage liver disease; HCC: hepatocellular carcinoma.

**Table 5 cancers-13-02554-t005:** Predictors of microvascular invasion or a high grade of differentiation (G3 or G4) using data available before a liver resection.

Variables	Univariate Analysis		
	HR	95% CI	*p*-Value
Sex (M/F)			0.972
Age ≥ 70 years			0.255
BMI ≥ 30 kg/m^2^			0.929
Diabetes mellitus			0.282
HCV Ab+, n (%) HCV RNA+, n (%)			0.559 0.999
HBV S Ag+, n (%)			0.253
MELD ≥ 9			0.168
AFP (ng/mL)			0.535
AFP ≥ 200 ng/mL			0.337
Liver stiffness > 14 kPa			0.140
Preoperative tumor size			0.154
Preoperative tumor size ≥30 mm			0.469
TLRpeak60 p50	7.50	1.87–29.98	0.004

BMI: body mass index; HCV: hepatitis C virus; HBV: hepatitis B virus; MELD: model for end-stage liver disease; AFP: alpha-fetoprotein.

## Data Availability

The data used and analyzed in the current study will be provided by the corresponding author upon reasonable request.

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
