# Peer review of "2-[18F]FDG PET/CT as a Predictor of Microvascular Invasion and High Histological Grade in Patients with Hepatocellular Carcinoma"

_cancers, 2021, doi:10.3390/cancers13112554_

Round 1
Reviewer 1 Report
I thank the authors for their work in revising the manuscript according to previous suggestions.
This article gives some novel information on the role of FDG uptake in HCC patients. Although the results need to be confirmed, PET-CT could be a useful tool for predicting prognosis and helping selection of therapies.
I recommend the manuscript for publication, and I only suggest changing the conclusion sentences as follows:
Abstract “…semiquantitative analysis of certain metabolic parameters on PET/CT may help…”
Conclusions “…routine performance of a PET/CT previous to a HCC resection may be useful…”
Reviewer 2 Report
My major concern is the short follow-up period for recurrence of HCC. The analysis of prognostic factors for tumor recurrence after liver resection (Table 2) is based on the mean follow-up of only 8.0 months (29.3% at 12 months). Since the authors described high recurrence rate (70%) after resection at 5 year in Introduction (ref.3), the observation period of the study should be longer for the detection of main outcome for analysis. Under the high rate of recurrence with appropriate long observation period, the results for analysis might be changed.
Reviewer 3 Report
The authors responded to all the reviewers comments and the manuscript can now be accepted.
This manuscript is a resubmission of an earlier submission. The following is a list of the peer review reports and author responses from that submission.
Round 1
Reviewer 1 Report
This is an interesting study on FDG uptake in HCC patients undergoing surgical resection.
In section 2.5 it is reported a minimum follow-up period of 10 years (line 148) that is not correct.
There is a very limited number of patients (n=41) with a short follow-up period (8.0±5.2 months). The overall accuracy of the investigated parameters (SULpeak60, TLRpeak60, COMBIpeak60) is slightly fair (AUC 0.716, 0.744, and 0.71, respectively).
The role of preoperative FDG uptake as predictor of microvascular invasion in HCC has been already investigated (e.g. Hyun SH 2018, Kobayashi T 2016, Ahn SY 2015, Kornberg A 2009), this issue should be expanded. Also, the role of other imaging modalities should be discussed (e.g. Huang J 2020, Mulé S 2020, Zhang Y 2021).
There is a recent review (Lee SM 2019) on the clinical value of FDG PET in HCC management that should be included in the discussion.
Thera are e few typing errors throughout the text.
Reviewer 2 Report
My major concern is the short follow-up period for recurrence of HCC. The analysis of prognostic factors for tumor recurrence after liver resection (Table 2) is based on the mean follow-up of only 8.0 months (29.3% at 12 months). Since the authors described high recurrence rate (70%) after resection at 5 year in Introduction (ref.3), the observation period of the study should be longer for the detection of main outcome for analysis. Under the high rate of recurrence with appropriate long observation period, the results for analysis might be changed.
Reviewer 3 Report
The author report on correlation of PET variables with outcome of hepatocellular carinoma in a prospective patient cohort. The article is of interest and well written. I have only a few minor comments.
- Methods: "Patients included in this study were diagnosed with HCC according to the criteria of the established guidelines" please briefly summarize the patient inclusion criteria if of interest, e.g. a particular tumor stage, extension, histologically proven, etc.
- Methods: "the area under the curve using the three percentiles (p25, p50 and p75) was calculated." please explain, as it this is not very clear. Which quantity do these percentiles refer to?
- In the Discussion, there are sentences that repeat the results, e.g. "Among the analysed parameters...". Consider condensing these as much as possible.
- it is not very clear how the AUC could be calculated for a defined cut-off value, as AUC is the area of the ROC curve which respresents sensitivity/specificity for varying thresholds. Please clarify.
- Also, it is recommended providing
- Please provide evidence that the two variables, SULpeak60 and TLRpeak60 combined in the signature are indipendent (not intercorrelated)
- Standardized uptake value is usually abbreviated as SUV not SUL
- "bad prognosis"-->"poor prognosis"
- The conclusion that "cirrhotics with a high risk of having histological factors of worse prognosis could be directly selected for liver transplantation, or treated with transarterial chemoembolization as a neoadjuvant therapy prior to tumor resection." seems not correlated with the results. The results do not involve cirrhosis, so this seems too much of an unproven hypothesis and should perhaps fit in the discussion, not in the conclusion.